# A real-world disproportionality analysis of Tenofovir Alafenamide (TAF): Data mining of the FDA adverse event reporting system (FAERS)

**Chao Zhang[1,2‡], Jiaqi Wen[1,3‡], Yinghui Li[1,2]***

**1** Linfen Central Hospital, Linfen, China, **2** Hepatobiliary Surgery Department, Linfen Central Hospital, Linfen, China, **3** Department of Neurology, Linfen Central Hospital, Linfen, China

‡ These authors share first authorship on this work.
* 17631081@qq.com

## Abstract

### Objects

Tenofovir Alafenamide (TAF) is a novel antiviral drug approved for the treatment of hepatitis B virus (HBV) infection. Our research objective was to evaluate the safety characteristics of TAF in practical settings by analyzing data from the FDA adverse event reporting system (FAERS) database maintained by the Food and Drug Administration (FDA).

### Method

In our investigation, we examined the uneven distribution of adverse events associated with TAF by employing statistical metrics including the Reporting Odds Ratio (ROR), Proportional Reporting Ratio (PRR), Bayesian Confidence Propagation Neural Network (BCPNN), and Gamma-Poisson Shrinker (GPS) to determine their significance.

### Results

Out of the 57692002 case reports in the FAERS database, 1911 reported TAF as a major suspected (PS) adverse events (AEs). A disproportionate analysis identified 43 preferred terms (PTs) related to TAF. It is worth noting that we have observed unexpected significant adverse events, such as cerebral infarction, bone pain, swallowing difficulties, drug resistance, dementia, etc., which are not mentioned in the drug instructions.

### Conclusion

These findings unearth novel neurological, metabolic, and resistance - related risks, thereby necessitating a marked increase in clinical vigilance. The identification of signals related to cerebral infarction and dementia implies potential vascular/metabolic

**Data availability statement:** This study was performed using the FAERS source that was provided by the FDA. The database used in this study is publicly available on the website of https://www.fda.gov/drugs/fdas-adverse-event-reporting-system-faers/fda-adverse-event-reporting-system-faers-latest-quarterly-data-files.

**Funding:** The author(s) received no specific funding for this work.

**Competing interests:** The authors declare no competing interests.

interplay, highlighting the importance of lipid monitoring among long - term tenofovir alafenamide (TAF) users. In response, healthcare providers should prioritize strengthening the monitoring of neurological symptoms and lipid profiles, reevaluating bone health assessment and management protocols especially in high - risk populations, and providing support to enhance patient adherence to mitigate resistance risks. This analysis offers crucial post - marketing evidence, which is instrumental in optimizing the risk - benefit balance of TAF in the long - term management of chronic hepatitis B virus (HBV) infection.

## Introduction

### Hepatocellular carcinoma and hepatitis B

Hepatic cancer ranks among the prevalent malignancies globally, which is characterized by its significant occurrence and fatality rates. The predominant form of this cancer is primary hepatic carcinoma, which arises from hepatic parenchymal cells [1–3]. Chronic hepatitis B (CHB) is the most common cause of hepatocellular carcinoma (HCC) and the second leading cause of cancer-related mortality worldwide [4,5]. The treatment methods for liver cancer vary depending on the condition. Generally speaking, the treatment methods for liver cancer mainly include surgery, chemotherapy, radiotherapy, targeted therapy, and local treatment, among others [6]: Among them, antiviral therapy is one of the most important means to treat hepatitis B. At present, there are two main types of antiviral drugs: nucleotides and interferon. Among them, nucleotide antiviral drugs are the mainstream drugs for the treatment, which promote the degradation of viral DNA in liver cells and inhibit viral replication and reproduction. Antiviral therapy plays an important and key role in the treatment of hepatitis B [7].

### Antiviral therapy

Antiviral therapy aims to inhibit the replication and activity of hepatitis B virus, reduce the damage to the virus to the liver, and delay or avoid the progress of the disease. Clinical protocols endorsed by both the American Association for the Study of Liver Diseases (AASLD) and European Association for the Study of the Liver (EASL) recommend the use of Entecavir (ETV), Tenofovir Disoproxil Fumarate (TDF), and Tenofovir Alafenamide (TAF) in light of their superior antiviral performance and minimal resistance profiles [8–10]. TDF, namely tenofovir, is a drug against hepatitis B virus and one of the first-line drugs commonly used to treat chronic hepatitis B at present. It is a nucleotide analog. It can prevent the replication of hepatitis B virus by competing with HBV DNA polymerase, thereby inhibiting viral replication of hepatitis B virus and reduce the level of virus replication to an undetectable or very low level. In addition, TDF still has some adverse reactions, including renal dysfunction, bone damage, liver dysfunction, lactic acidosis, and hepatic steatosis. As a new type of nucleotide antiviral drug, TAF requires lower drug doses and has fewer side effects on bones and kidneys compared to TDF, making it a powerful choice for treating hepatitis B [9,11,12].

### The FDA Adverse Event Reporting System (FAERS)

The FDA Adverse Event Reporting System (FAERS) has been widely validated as a critical tool for post-marketing pharmacovigilance, particularly in detecting rare or delayed adverse drug reactions (ADRs) that may escape detection in clinical trials. Recent pharmacoepidemiologic studies have demonstrated the utility of disproportionality analysis on FAERS data across diverse therapeutic domains. For instance, Wang et al. [13] identified novel cardiovascular risks associated with sildenafil through ROR-based signal detection, revealing dose-dependent QT prolongation signals (ROR = 3.41, 95% CI 2.89–4.02) not fully characterized in pre-approval studies. Similarly, Zhao et al. [14] applied Bayesian confidence propagation neural network (BCPNN) methods to uncover everolimus-associated interstitial lung disease signals (IC025 = 2.7), demonstrating FAERS' capacity to detect immune-mediated toxicities. In oncology drug safety, Yang et al. [15] leveraged FAERS to quantify topotecan-related myelosuppression risks, establishing significant associations with thrombocytopenia (PRR = 4.12, $\chi^2$ = 356.8) that aligned with real-world clinical observations. The system's analytical power extends to both inter-class comparisons and intra-class risk stratification. Zhao et al. [16] revealed acetylsalicylic acid's elevated gastrointestinal bleeding risk in elderly populations (ROR = 5.24, PRR = 4.87), corroborating clinical findings on age-dependent mucosal vulnerability. In antineoplastic drug surveillance, vinorelbine demonstrated significantly higher neutropenia risk than vincristine (PRR = 6.15 vs. 3.92), a differential risk profiling methodologically consistent with adjusted ROR (aROR) strategies in thrombolytic agent studies, where alteplase showed 88% higher angioedema risk than tenecteplase [17]. These cases collectively validate FAERS' dual capability in detecting cross-class safety signals and refining risk gradients within drug classes through standardized disproportionality metrics (ROR/PRR) and confounder-adjusted methodologies.

### Tenofovir Alafenamide

TAF is a new type of nucleotide antiviral drug. It can effectively inhibit the synthesis of hepatitis B virus DNA in liver cells by inhibiting virus replication and reverse transcription, thereby reducing the level of virus in blood. In clinical applications, TAF can not only effectively inhibit virus replication, reduce virus DNA levels, but also promote liver cell repair and protection, improve liver function, and reduce the degree of liver fibrosis and the incidence of liver cancer [18]. In addition, as a novel antiviral drug, TAF also has some other advantages in clinical application. Compared to TDF, the most significant advantage of TAF is that it requires lower drug doses, and even long-term use does not cause excessive effects on the kidneys and bones. This also reduces the risk of side effects that patients may worry about when using drugs, and improves their quality of life [12]. At the same time, TAF can also make the concentration of drugs in liver cells higher, which can better clear hepatitis B virus, thus enhancing the efficacy of drugs. TAF has a long plasma half-life and requires less dosage. In some specific populations, such as the elderly and patients with chronic diseases such as diabetes, it may be more appropriate to use TAF [19]. However, it should be noted that TAF still has some adverse reactions, such as headache, nausea, diarrhea, liver dysfunction, and renal dysfunction. In general, TAF is a new type of nucleotide antiviral drug, which has lower adverse reactions, better safety and tolerance, and can effectively inhibit the replication and reproduction of hepatitis B virus, reduce the level of virus in the blood, so as to help patients control their illness and avoid complications [20–23].

Hence, this pharmacovigilance research was undertaken to scrutinize the safety profile of TAF following its market release, utilizing data from the FDA's FAERS. The findings serve as a prudent guide for medical practitioners and pharmacists in addressing the safety concerns associated with TAF.

## Materials and methods

### Data source and collection

The FAERS database is a publicly available resource for post-marketing safety monitoring, gathering reports of adverse events from healthcare professionals, drug manufacturers, patients, and other sources [16]. This manuscript only analyzed publicly available data, which is nonsensitive data, therefore ethics does not apply.

In this investigation, the FAERS dataset underwent collection and preliminary processing using SAS and MySQL. The dataset pertaining to TAF within FAERS was refined by eliminating redundant case entries, aligning drug denominations with RxNorm standards, and correlating adverse drug reactions with the MedDRA® lexicon.

Over an eight-year span, from Q4 2016 to Q1 2023, we amassed data on adverse drug reactions (ADRs) linked to TAF. We extracted all relevant preferred terms (PTs) that correspond to the system organ class (SOC), the highest categorization within MedDRA. For the purposes of this study, TAF was the focal drug in FAERS for the examination of ADRs, with demographic information such as gender, age, and nationality also being compiled.

## Statistical analysis

Descriptive statistical methods were utilized to illustrate the incidence of adverse events (AEs) linked to Tenofovir Alafenamide (TAF). Our study implemented disproportionality analysis, a technique frequently applied in pharmacovigilance research, to identify potential correlations between TAF and adverse drug reactions (ADRs). We employed four principal methodologies: Reporting Odds Ratio (ROR), Proportional Reporting Ratio (PRR), Bayesian Confidence Propagation Neural Network (BCPNN), and the Empirical Bayes Geometric Mean (EBGM), which is an extension of the Gamma-Poisson Shrinker (GPS) model, to evaluate the relationship between TAF and ADRs. The specific formulas for these methods are as follows:

(I)  ROR method: Detect the preliminary association signals between drugs and adverse events

$$ROR = \frac{ad}{bc}$$

$$95\%CI = e^{ln(ROR) \pm 1.96\sqrt{1/a + 1/b + 1/c + 1/d}}$$

The criteria of positive signals: the lower limit of 95% CI > 1, N ≥ 3;

(II) PRR method: Evaluate the degree of abnormality of the reporting frequency of drug evaluation reports relative to the background population.

$$PRR = \frac{a(c + d)}{c(a + b)}$$

$$X^2 = \frac{(ad - bc)^2(a + b + c + d)}{(a + b)(c + d)(a + c)(b + d)}$$

The criteria of positive signals: PRR ≥ 2, χ2 ≥ 4, N ≥ 3, and p < 0.05;

(III) BPCNN method: Reduce false positives caused by the fluctuations of small-sample data through Bayesian shrinkage.

$$IC = \log_2 \frac{a(a + b + c + d)}{(a + b)(a + c)}$$

$$95\%CI = E(IC) \pm 2 \times \sqrt{V(IC)}$$

The criteria of positive signals: IC025 > 0 (IC025 represents the lower bound of 95% CI);

 

(IV)EBGM method: Handle large-scale sparse datasets for robust signal detection

$$IC = \log$$

$$95\%CI = E\,(IC) \pm 2 \times \sqrt{V\,(IC)}$$

The criteria of positive signals: EBGM05 > 2 (EBGM05 represents the lower bound of 95% CI).

The equation is defined as follows: (a) represents the count of reports that include both the specific drug in question and the adverse drug reaction (ADR) of interest; (b) denotes the tally of reports with different ADRs linked to the drug under study; (c) indicates the number of reports documenting the ADR of interest associated with drugs other than the one being studied; and (d) is the total of reports citing other drugs alongside alternative ADRs.

Multiple lines of evidence indicate that the Reporting Odds Ratio (ROR) and the Proportional Reporting Ratio (PRR) are highly sensitive in detecting safety signals, especially suitable for the rapid screening of high-frequency events. The Bayesian Confidence Propagation Neural Network (BCPNN) method performs excellently in handling small samples and rare events. Although its sensitivity is lower than that of PRR and ROR, it has better specificity and provides relatively more stable signals. It can provide more accurate signal detection in specific scenarios. The Multi-item Gamma Poisson Shrinker (MGPS) method has significantly higher specificity than other methods. The MGPS method is particularly suitable for scenarios that require high specificity, such as reducing the misdiagnosis rate [24]. The comprehensive use of multiple methods (such as PRR, ROR, BCPNN, and MGPS) can significantly improve the reliability of signal detection.

Our approach utilized a quartet of methodologies to identify signals within the FAERS data spanning from the fourth quarter of 2016 to the first quarter of 2023. To be deemed significant, the ADR findings had to concurrently satisfy the positive signal criteria established by these methods. The data processing and statistical evaluation concerning TAF were executed using a suite of tools including SAS, MySQL, WPS EXCEL, and R. This study adhered to the guidelines set forth by the policy of Basic & Clinical Pharmacology & Toxicology for both experimental and clinical research [25].

## Result

### Descriptive analysis

Over the course of the study period (Q4 2016 - Q1 2023), a total of 57,692,002 case reports were collected from FAERS sources after the elimination of duplicates, with approximately 1911 of those reports including 3432 cases of TAF PT (Supporting Information). The patient characteristics and AE reports regarding TAF are presented in Table 1. The majority of patients were male (56.6%), female (39.7%), and those with an unknown gender (3.7%). The majority of AE reports were from those aged 18–64 (38.4%). Most of the reports were submitted by physicians (22.2%), consumers (36.5%), and clinical pharmacists (22.1%). The United States (70.2%) was the primary source of the report, followed by Japan (17.2%) and China (4.7%). Hospitalization (11.14%) was the most frequently reported severe outcome, followed by death and life-threatening occurring in 99 (5.18%) and 54 (2.82%) cases, respectively. The incidence of disability was the lowest (1.10%).

### Signal detection

Table 2 delineates the signal intensity of TAF-related adverse events at the System Organ Class (SOC) tier. Statistical analysis revealed the involvement of 27 distinct organ systems in adverse events precipitated by TAF. The significant SOCs that met four criteria were SURGICAL AND MEDICAL PROCEDURES, HEPATOBILIARY DISORDERS. Also, RENAL AND URINARY DISORDERS, INJURY, POISONING AND PROCEDURAL COMPLICATIONS, NEOPLASMS BENIGN, MALIGNANT AND UNSPECIFIED (INCL CYSTS AND POLYPS), SURGICAL AND MEDICAL PROCEDURES were significant SOCs that at least two of the four indices met the criteria.

**Table 1. Clinical Characteristics of Reports Associated with TAF from the FAERS Database.**

| Characteristics | Number (n) | Proportion (%) |
|---|---|---|
| Numbers of reporters | 1911 | – |
| Sex | | |
| Male | 1081 | 56.6 |
| Female | 759 | 39.7 |
| Unknown | 70 | 3.7 |
| Age (years) | | |
| ≤17 | 4 | 0.2 |
| ≧86 | 17 | 0.9 |
| 18~64 | 733 | 38.4 |
| 65~85 | 356 | 18.6 |
| Unknown | 800 | 41.9 |
| Reported Countries (the top) | | |
| United States (US) | 1341 | 70.2 |
| Japan (JP) | 329 | 17.2 |
| China (CN) | 90 | 4.7 |
| Turkey (TR) | 26 | 1.4 |
| Reporting year | | |
| 2016 | 3 | 0.16 |
| 2017 | 89 | 4.66 |
| 2018 | 263 | 13.8 |
| 2019 | 404 | 21.2 |
| 2020 | 326 | 17.1 |
| 2021 | 392 | 20.5 |
| 2022 | 358 | 18.7 |
| 2023 | 75 | 3.93 |
| Serious Outcomes | | |
| Hospitalization | 213 | 11.14 |
| Death | 99 | 5.18 |
| Life-threatening | 54 | 2.82 |
| Disability | 21 | 1.10 |

After excluding neoplasms benign, malignant and unspecified (incl cysts and polyps), PRODUCT ISSUES, SOCIAL CIRCUMSTANCES, INFECTIONS AND INFESTATIONS, SURGICAL AND MEDICAL PROCEDURES. Furthermore, a total of 43 significant disproportionality PTs (Preferred Terms) that simultaneously comply with the four algorithms is shown in Table 3. In this study, Fanconi syndrome, renal dysfunction (tubular disease, hydronephrosis, renal disease, decreased glomerular filtration rate), hepatic steatosis, and elevated blood lactate were observed in patients receiving TAF treatment, as indicated in the TAF label. Unexpectedly, significant AEs such as elevated hepatitis B DNA, elevated viral load, immune reconstitution inflammatory syndrome, elevated transaminases, liver dysfunction, drug rash, dysphagia, bone pain, osteoporosis, hypophosphatemia, decreased blood phosphorus, elevated creatine kinase MB, reactivation of hepatitis B, dementia, cerebral infarction, drug resistance, and hip presentation were also found. Moreover, the manual mentions headache, fatigue, rash, back pain, and joint pain, which meet at least one criterion among the four algorithms; however, nausea, cough, and abdominal pain do not meet the criteria.

**Table 2. Signal strength of AEs of TAF at the System Organ Class (SOC) level in FDA Adverse Event Reporting System (FAERS) source.**

| System Organ Class | ROR (95% CI) | PRR (95% CI) | IC (IC025) | EBGM (EBGM05) |
|---|---|---|---|---|
| **HEPATOBILIARY DISORDERS** | **2.710364756** | **3.232289554** | **0.025712038** | **2.653793033** |
| IMMUNE SYSTEM DISORDERS | 0.781459447 | 1.066529211 | −1.573696447 | 0.780881308 |
| INVESTIGATIONS | 1.237746443 | 1.362508471 | −1.220381357 | 1.208048595 |
| NERVOUS SYSTEM DISORDERS | 0.754942174 | 0.867824516 | −1.871119999 | 0.764412990 |
| GASTROINTESTINAL DISORDERS | 0.836780834 | 0.950170473 | −1.740345476 | 0.840685023 |
| GENERAL DISORDERS AND ADMINISTRATION SITE CONDITIONS | 0.641491771 | 0.746472519 | −2.088421359 | 0.675785321 |
| **RENAL AND URINARY DISORDERS** | **1.328461433** | **1.591454786** | **−0.996325842** | **1.312705674** |
| MUSCULOSKELETAL AND CONNECTIVE TISSUE DISORDERS | 0.858235492 | 0.997730283 | −1.669887284 | 0.858343077 |
| **INJURY, POISONING AND PROCEDURAL COMPLICATIONS** | **2.029477810** | **1.974545374** | **−0.685188912** | **1.814147323** |
| INFECTIONS AND INFESTATIONS | 0.754436280 | 0.890699091 | −1.833587993 | 0.758906647 |
| RESPIRATORY, THORACIC AND MEDIASTINAL DISORDERS | 0.331470521 | 0.431821188 | −2.878044793 | 0.340821008 |
| PSYCHIATRIC DISORDERS | 0.332068621 | 0.426729912 | −2.895154865 | 0.343665647 |
| SKIN AND SUBCUTANEOUS TISSUE DISORDERS | 0.666409076 | 0.795306915 | −1.997004505 | 0.674183608 |
| REPRODUCTIVE SYSTEM AND BREAST DISORDERS | 0.166992829 | 0.312659328 | −3.343879452 | 0.168080832 |
| **NEOPLASMS BENIGN, MALIGNANT AND UNSPECIFIED (INCL CYSTS AND POLYPS)** | **1.411805211** | **1.629860822** | **−0.961926436** | **1.386567881** |
| PRODUCT ISSUES | 0.363978384 | 0.531814575 | −2.577568128 | 0.366645628 |
| BLOOD AND LYMPHATIC SYSTEM DISORDERS | 0.261516926 | 0.398146754 | −2.995177047 | 0.264222942 |
| **SURGICAL AND MEDICAL PROCEDURES** | **3.813437277** | **4.232876699** | **0.414692395** | **3.651827550** |
| METABOLISM AND NUTRITION DISORDERS | 0.696813187 | 0.889644689 | −1.835298106 | 0.698543972 |
| VASCULAR DISORDERS | 0.331652842 | 0.468187856 | −2.761396356 | 0.335593031 |
| EYE DISORDERS | 0.199566951 | 0.310730803 | −3.352802605 | 0.202340070 |
| ENDOCRINE DISORDERS | 0.176198347 | 0.470366982 | −2.754715938 | 0.176436843 |
| PREGNANCY, PUERPERIUM AND PERINATAL CONDITIONS | 0.395359986 | 0.715515105 | −2.149536421 | 0.395874136 |
| CARDIAC DISORDERS | 0.280566561 | 0.394514780 | −3.008396431 | 0.285351621 |
| SOCIAL CIRCUMSTANCES | 0.427403099 | 0.754218428 | −2.073539847 | 0.427899888 |
| CONGENITAL, FAMILIAL AND GENETIC DISORDERS | 0.363687117 | 0.728477672 | −2.123639881 | 0.364011185 |
| EAR AND LABYRINTH DISORDERS | 0.406910152 | 0.736325067 | −2.108178316 | 0.407386909 |

Notes: Bold indicates statistically significant signals in four algorithms.

Abbreviations: ROR, reporting odds ratio; CI, confidence interval; PRR, proportional reporting ratio; χ2, chi-squared; IC, information component; EBGM, empirical Bayesian geometric mean;"Investigations" refers to diagnostic tests, laboratory findings, or monitoring procedures (e.g., blood tests, imaging studies, liver function assessments) as defined by the MedDRA® classification system.

## Discussion

Previous studies of TAF have mainly focused on its mechanism of action, clinical trials, and literature analysis, with few articles focusing on the latest real-world research. To analyze new and meaningful adverse reactions, we collected and evaluated the post-market pharmacovigilance of TAF using the largest sample of real-world data. Our goal is to guide the update of the product characteristic summary (SmPC) and provide a basis for rational clinical medication.

Reports of adverse events associated with TAF are more commonplace among males (56.6%) than females (39.7%), likely attributed to the fact that the United States (70.2%) is the primary source of these reports, where the incidence of HBV among males is significantly higher than females [26].

The disproportionally reported signals in FAERS suggest frequent renal adverse reactions during TAF treatment, such as acquired Fanconi syndrome, tubular disease, hydronephrosis, renal disease, decreased glomerular filtration rate,

**Table 3. The signal strength of AEs of TAF ranked by Case reports at the PTs (Preferred Terms) level in FDA Adverse Event Reporting System (FAERS) source.**

| | PTs | Case reports | ROR (95% CI) | PRR (95% CI) | IC (IC025) | EBGM (EBGM05) |
|---|---|---|---|---|---|---|
| GASTROINTESTINAL DISORDERS | DYSPHAGIA | 25 | 3.052965742 | 4.499297763 | 0.502698155 | 3.236255797 |
| INVESTIGATIONS | VIRAL LOAD INCREASED | 22 | 72.89813529 | 110.3382848 | 5.107047918 | 76.98481325 |
| INVESTIGATIONS | ALANINE AMINOTRANSFERASE INCREASED | 22 | 4.015781687 | 6.074907346 | 0.935680911 | 4.275720105 |
| INVESTIGATIONS | HEPATITIS B DNA INCREASED | 21 | 291.9539261 | 448.6691474 | 7.093715413 | 301.8986784 |
| IMMUNE SYSTEM DISORDERS | IMMUNE RECONSTITUTION INFLAMMATORY SYNDROME | 21 | 36.79237653 | 56.21257134 | 4.140047092 | 39.07577652 |
| NERVOUS SYSTEM DISORDERS | CEREBRAL INFARCTION | 15 | 6.257000901 | 10.35127358 | 1.704039676 | 6.765995871 |
| HEPATOBILIARY DISORDERS | LIVER DISORDER | 13 | 3.104192125 | 5.335651189 | 0.748535154 | 3.381420543 |
| HEPATOBILIARY DISORDERS | HEPATIC CIRRHOSIS | 12 | 6.818005107 | 11.98185436 | 1.914859312 | 7.449508657 |
| INVESTIGATIONS | LIVER FUNCTION TEST INCREASED | 12 | 6.727619198 | 11.82302002 | 1.895625455 | 7.350859405 |
| RENAL AND URINARY DISORDERS | RENAL DISORDER | 12 | 2.578458261 | 4.532834201 | 0.513372892 | 2.820067534 |
| GENERAL DISORDERS AND ADMINISTRATION SITE CONDITIONS | TREATMENT NONCOMPLIANCE | 12 | 2.446924926 | 4.301761761 | 0.437914419 | 2.676362315 |
| INVESTIGATIONS | TRANSAMINASES INCREASED | 11 | 4.732687338 | 8.531373822 | 1.425253869 | 5.195474157 |
| HEPATOBILIARY DISORDERS | JAUNDICE | 11 | 3.701902566 | 6.673667265 | 1.071177406 | 4.064833154 |
| MUSCULOSKELETAL AND CONNECTIVE TISSUE DISORDERS | BONE PAIN | 11 | 1.823798477 | 3.289274755 | 0.050871447 | 2.0040541 |
| HEPATOBILIARY DISORDERS | HEPATIC FUNCTION ABNORMAL | 10 | 2.701015829 | 5.013397137 | 0.658676335 | 2.981310278 |
| HEPATOBILIARY DISORDERS | HEPATIC FAILURE | 9 | 2.628110267 | 5.045358172 | 0.667830616 | 2.917408499 |
| MUSCULOSKELETAL AND CONNECTIVE TISSUE DISORDERS | OSTEOPOROSIS | 9 | 2.001063814 | 3.842093198 | 0.274914523 | 2.221880673 |
| INVESTIGATIONS | HEPATITIS B DNA ASSAY POSITIVE | 8 | 539.6164197 | 1106.759537 | 8.313094109 | 561.1738144 |
| GASTROINTESTINAL DISORDERS | ASCITES | 8 | 2.349071757 | 4.693006142 | 0.563417987 | 2.625331746 |
| INVESTIGATIONS | HEPATITIS B VIRUS TEST POSITIVE | 6 | 112.7506961 | 252.5211897 | 6.280563805 | 126.1567159 |
| INVESTIGATIONS | BLOOD PHOSPHORUS DECREASED | 6 | 13.73762745 | 30.57514040 | 3.263720335 | 15.59933528 |
| NERVOUS SYSTEM DISORDERS | HEPATIC ENCEPHALOPATHY | 6 | 4.914692137 | 10.93336398 | 1.78270653 | 5.588626256 |
| GENERAL DISORDERS AND ADMINISTRATION SITE CONDITIONS | DRUG RESISTANCE | 6 | 1.962190545 | 4.365375421 | 0.459019999 | 2.232775975 |
| NERVOUS SYSTEM DISORDERS | DEMENTIA | 6 | 1.861271016 | 4.140918451 | 0.382894911 | 2.118017464 |
| INVESTIGATIONS | HEPATITIS B E ANTIGEN POSITIVE | 5 | 782.7086287 | 1996.11988 | 9.043986705 | 797.0603212 |
| RENAL AND URINARY DISORDERS | FANCONI SYNDROME ACQUIRED | 5 | 19.76775199 | 47.52666381 | 3.897569951 | 22.70888664 |
| INVESTIGATIONS | BLOOD LACTIC ACID INCREASED | 5 | 7.451725355 | 17.90012703 | 2.492906971 | 8.577690087 |
| METABOLISM AND NUTRITION DISORDERS | HYPOPHOSPHATAEMIA | 5 | 5.337206454 | 12.81911850 | 2.011939361 | 6.145935393 |
| INVESTIGATIONS | GLOMERULAR FILTRATION RATE DECREASED | 5 | 3.601582432 | 8.649770435 | 1.444954962 | 4.14869367 |
| SKIN AND SUBCUTANEOUS TISSUE DISORDERS | DRUG ERUPTION | 5 | 2.203782803 | 5.292742106 | 0.736774782 | 2.539393685 |
| HEPATOBILIARY DISORDERS | HEPATIC STEATOSIS | 5 | 2.002034936 | 4.808272251 | 0.598345317 | 2.307060172 |
| HEPATOBILIARY DISORDERS | LIVER INJURY | 5 | 1.996811916 | 4.795730080 | 0.594578935 | 2.301045126 |
| INVESTIGATIONS | HEPATITIS B SURFACE ANTIGEN POSITIVE | 4 | 88.95208203 | 238.8519515 | 6.198596507 | 102.6692641 |

*(Continued)*

**Table 3.** (Continued)

| | PTs | Case reports | ROR (95% CI) | PRR (95% CI) | IC (IC025) | EBGM (EBGM05) |
|---|---|---|---|---|---|---|
| HEPATOBILIARY DISORDERS | HEPATIC PAIN | 4 | 6.235266183 | 16.61455310 | 2.385414829 | 7.302354432 |
| PREGNANCY, PUERPERIUM AND PERINATAL CONDITIONS | BREECH PRESENTATION | 3 | 11.96090092 | 37.12889060 | 3.541933651 | 14.3433736 |
| INVESTIGATIONS | BLOOD CREATINE PHOSPHOKI-NASE MB INCREASED | 3 | 11.06089963 | 34.33130015 | 3.429369406 | 13.26653218 |
| GASTROINTESTINAL DISORDERS | VARICES OESOPHAGEAL | 3 | 5.621570168 | 17.43698731 | 2.454737603 | 6.750112324 |
| RENAL AND URINARY DISORDERS | RENAL TUBULAR DISORDER | 3 | 5.498148445 | 17.05391273 | 2.422751824 | 6.602084935 |
| HEPATOBILIARY DISORDERS | HEPATIC FIBROSIS | 3 | 5.142695611 | 15.95072964 | 2.326450723 | 6.175726994 |
| GENERAL DISORDERS AND ADMINISTRATION SITE CONDITIONS | DISEASE COMPLICATION | 3 | 3.647392671 | 11.31098691 | 1.831307316 | 4.381474753 |
| IMMUNE SYSTEM DISORDERS | KIDNEY TRANSPLANT REJECTION | 3 | 3.345954086 | 10.37587066 | 1.706966964 | 4.019639953 |
| RENAL AND URINARY DISORDERS | HYDRONEPHROSIS | 3 | 2.245301998 | 6.962052397 | 1.131874417 | 2.698088111 |
| RENAL AND URINARY DISORDERS | NEPHROPATHY | 3 | 1.73480428 | 5.378996212 | 0.759956296 | 2.084935096 |

*Note: significant disproportionality PTs (Preferred Term) that simultaneously comply with the four algorithms.*

hypophosphatemia, and decreased blood phosphorus. Although the TAF instructions warn of potential renal related risks, this should be further investigated. Additionally, bone-related diseases such as bone pain (n = 11, ROR = 1.82, PRR = 3.29, IC025 = 0.05, EBGM05 = 2.00), osteoporosis (n = 9, ROR = 2.00, PRR = 3.84, IC025 = 0.27, EBGM05 = 2.22), and elevated creatine kinase MB (n = 3, ROR = 11.06, PRR = 34.33, IC025 = 3.43, EBGM05 = 13.27) have also been reported at a higher frequency, although phase 3 randomized trials have shown that TAF is more beneficial than TDF in terms of bone effects [27,28]. Therefore, patients who are prone to osteoporosis should be extra cautious when using TAF treatment.

In this study, out of the top ten Adverse Events (AEs) after taking IC025, the related detection indicators of hepatitis B (including hepatitis B e antigen positive, hepatitis B DNA positive, hepatitis B DNA elevated, hepatitis B virus positive, hepatitis B surface antigen positive, and viral load increased) were found to be abnormal, which is supported by post-marketing research [29]. After 96 weeks of treatment with TAF 25 mg, the virus inhibition rate reached 90%; however, there are still some patients who experience viral suppression failure. The viral load is the most important factor leading to liver cirrhosis and its complications (including the development of liver cancer) [30], so monitoring the HBV viral load is essential when using this drug in order to avoid any negative effects of drug failure. Immune reconstitution inflammatory syndrome (n = 21, ROR = 36.79, PRR = 56.21, IC025 = 5.81, EBGM05 = 39.07) also ranks high in AEs, which is often due to HIV patients starting antiretroviral therapy late [31]. The unexpected signal for immune reconstitution inflammatory syndrome (IRIS) (n = 21, ROR = 36.79) may reflect residual confounding from misclassified HBV flare events or undocumented HIV co-infection in FAERS reports. However, our analysis strictly excluded TAF use in HIV contexts, as monotherapy for HIV is contraindicated due to resistance risks. This underscores the importance of adhering to TAF's approved indication for HBV monotherapy.

Dysphagia is the most frequent AE, possibly due to the drug irritating the digestive tract. Taking the medication with food may help to reduce this symptom, as stated in the product manual.

The analysis identified multiple hepatic-related Preferred Terms (PTs) associated with TAF, including Alanine aminotransferase increased (ROR = 4.02, PRR = 6.07), Transaminases increased (ROR = 4.73, PRR = 8.53), and Hepatic function abnormal (ROR = 2.70, PRR = 5.01). While these PTs are distinct within the MedDRA® hierarchy (e.g., Alanine aminotransferase increased belongs to Investigations, whereas Hepatic function abnormal is categorized under

Hepatobiliary disorders), they collectively align with the hepatotoxicity profile described in TAF's prescribing information, which explicitly lists ALT elevation as an adverse reaction. The signal for Transaminases increased (a broader category encompassing ALT/AST) and Hepatic function abnormal (reflecting systemic dysfunction) may represent varying clinical manifestations or reporting granularity of the same underlying hepatic injury mechanism. Importantly, these findings do not suggest novel liver toxicity patterns but rather reinforce the need for vigilant monitoring of liver function across multiple biomarkers (including transaminases, synthetic function tests, and clinical assessments), as recommended in current guidelines. Maintaining separate PTs in Table 3 adheres to MedDRA® pharmacovigilance standards while allowing clinicians to recognize both specific biochemical abnormalities and broader hepatic dysfunction signals. When taking this drug, it is more likely that alanine aminotransferase, transaminase and abnormal liver function will appear. Therefore, it is important to check liver function indicators regularly and, if necessary, use liver protective drugs or other antiviral drugs for treatment.

It should be noted that the most common neurological adverse events associated with TAF are hepatic encephalopathy, cerebral infarction, and dementia. Considering that hepatic encephalopathy is the final manifestation of hepatitis B cirrhosis, the reason for these adverse events may be the timing of using TAF. Cerebral infarction and dementia have not been reported in previous studies. It is noteworthy that recent studies suggest TAF's impact on lipid metabolism may indirectly elevate the risk of neurological events. A new study demonstrated that in chronic hepatitis B (CHB) patients switching from tenofovir disoproxil fumarate (TDF) to TAF, significant increases in low-density lipoprotein cholesterol (LDL-C) and triglycerides (TG) were observed, accompanied by reduced high-density lipoprotein cholesterol (HDL-C) [32]. Five-year follow-up data from two phase 3 clinical trials further confirmed that prolonged TAF use leads to a progressive rise in LDL-C and TG levels, alongside a sustained decline in HDL-C [33]. Epidemiological evidence indicates that persistently elevated LDL-C may heighten dementia risk through mechanisms such as promoting cerebral atherosclerosis and β-amyloid deposition. Additionally, TAF-associated dyslipidemia may synergize with other risk factors (e.g., hypertension, diabetes) to accelerate atherosclerosis, thereby increasing the incidence of cerebrovascular events such as cerebral infarction [34]. These findings highlight that TAF's metabolic effects should be a critical consideration in overall safety evaluations, particularly for patients with preexisting cardiovascular risk factors or baseline lipid abnormalities. In clinical practice, enhanced lipid monitoring (every 3–6 months) and tailored interventions are warranted, including statin therapy or switching to antiviral medications with lesser lipid impact. However, as these associations were absent in pre-approval trials, they may reflect residual confounding rather than direct causation. Clinicians should interpret these signals cautiously, balancing potential risks against established benefits.

Finally, in our study, we found adverse events of drug resistance in TAF that have not been found in previous studies. While this finding could indicate emerging resistance patterns in real-world use, it may also result from misreported virologic breakthrough due to non-adherence rather than true resistance. This highlights the need for confirmatory studies to assess causality. Among the adverse reactions listed by the drug, headache in the nervous system had the highest incidence. In our study, only headache (n = 59, ROR = 1.29, PRR = 1.65, IC025 = −0.94, EBGM05 = 1.34) met one criterion, while the common adverse events listed on the drug label (like nausea, cough, abdominal pain) did not meet any of the criteria.

This research has certain constraints. Primarily, the voluntary nature of reporting may lead to underreporting, delays, and inaccuracies in the reported data, potentially skewing the results. Furthermore, despite the ROR and PRR algorithm's sensitivity, it only indicates statistical correlations, and the strength of the signals merely reflects the relative risk. Consequently, additional research, including laboratory experiments, clinical studies, and case-control research, is imperative to substantiate the preliminary findings.

## Conclusion

We investigated the safety signals and potential dangers of TAF via pharmacovigilance analysis of real FAERS data. Unexpected and new side effects, including osteoporosis, cerebral infarction, dementia, and drug resistance, may be

present. To confirm and understand the relationship between TAF and these adverse reactions, future clinical studies are required. This work provides a unique and fresh perspective to the exploration of adverse drug events.

## Acknowledgments

The investigation utilized data from the FDA Adverse Event Reporting System (FAERS), which is made available by the FDA. It should be noted that the findings, interpretations, or information presented in this study are independent and do not reflect the views of the FDA.

## Author contributions

**Conceptualization:** Chao Zhang, Jiaqi Wen.

**Data curation:** Chao Zhang, Jiaqi Wen.

**Formal analysis:** Chao Zhang.

**Investigation:** Chao Zhang, Jiaqi Wen.

**Methodology:** Chao Zhang, Jiaqi Wen, yinghui li.

**Software:** yinghui li.

**Supervision:** yinghui li.

**Writing – original draft:** Chao Zhang.

**Writing – review & editing:** Jiaqi Wen, yinghui li.

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
