## [Decision Letter · Decision Letter 0]

13 Feb 2025

Dear Dr. li,

Thank you for submitting your manuscript to PLOS ONE. After careful consideration, we feel that it has merit but does not fully meet PLOS ONE’s publication criteria as it currently stands. Therefore, we invite you to submit a revised version of the manuscript that addresses the points raised during the review process.

We look forward to receiving your revised manuscript.

Kind regards,

Douglas S. Krakower, MD

Academic Editor

PLOS ONE

Journal requirements:   When submitting your revision, we need you to address these additional requirements. 1. Please ensure that your manuscript meets PLOS ONE's style requirements, including those for file naming. The PLOS ONE style templates can be found at https://journals.plos.org/plosone/s/file?id=wjVg/PLOSOne_formatting_sample_main_body.pdf and https://journals.plos.org/plosone/s/file?id=ba62/PLOSOne_formatting_sample_title_authors_affiliations.pdf. 2. PLOS requires an ORCID iD for the corresponding author in Editorial Manager on papers submitted after December 6th, 2016. Please ensure that you have an ORCID iD and that it is validated in Editorial Manager. To do this, go to ‘Update my Information’ (in the upper left-hand corner of the main menu), and click on the Fetch/Validate link next to the ORCID field. This will take you to the ORCID site and allow you to create a new iD or authenticate a pre-existing iD in Editorial Manager. 3. We note that your Data Availability Statement is currently as follows: [All relevant data are within the manuscript and its Supporting Information files.] Please confirm at this time whether or not your submission contains all raw data required to replicate the results of your study. Authors must share the “minimal data set” for their submission. PLOS defines the minimal data set to consist of the data required to replicate all study findings reported in the article, as well as related metadata and methods (https://journals.plos.org/plosone/s/data-availability#loc-minimal-data-set-definition). For example, authors should submit the following data: - The values behind the means, standard deviations and other measures reported;- The values used to build graphs;- The points extracted from images for analysis. Authors do not need to submit their entire data set if only a portion of the data was used in the reported study. If your submission does not contain these data, please either upload them as Supporting Information files or deposit them to a stable, public repository and provide us with the relevant URLs, DOIs, or accession numbers. For a list of recommended repositories, please see https://journals.plos.org/plosone/s/recommended-repositories. If there are ethical or legal restrictions on sharing a de-identified data set, please explain them in detail (e.g., data contain potentially sensitive information, data are owned by a third-party organization, etc.) and who has imposed them (e.g., an ethics committee). Please also provide contact information for a data access committee, ethics committee, or other institutional body to which data requests may be sent. If data are owned by a third party, please indicate how others may request data access. 4. Please amend either the abstract on the online submission form (via Edit Submission) or the abstract in the manuscript so that they are identical.

Reviewers' comments:

Reviewer's Responses to Questions

**Comments to the Author**

1. Is the manuscript technically sound, and do the data support the conclusions?

Reviewer #1: Yes

Reviewer #2: Yes

2. Has the statistical analysis been performed appropriately and rigorously?

Reviewer #1: I Don't Know

Reviewer #2: Yes

3. Have the authors made all data underlying the findings in their manuscript fully available?

Reviewer #1: Yes

Reviewer #2: Yes

4. Is the manuscript presented in an intelligible fashion and written in standard English?

Reviewer #1: No

Reviewer #2: Yes

Reviewer #1: A disproportionality analysis was performed to investigate adverse event reporting associated with Tenofovir Alafenamide (TAF) in the FDA adverse event reporting system between 2016 and 2023. Four principal methodologies were used to detect signals in the database.

This is a straightforward analysis that offers useful hypothesis generating information for future studies. Some minor comments below:

Introduction: Includes a lot of run-on sentences and some grammatical errors. Authors should consider subheadings to break up sections to help the reader, as well as use a grammar checking software to restructure some awkward sentences. For example:

I. Hepatocellular Carcinoma

II. Hepatitis B (consider combining with section 1 and truncating - reader doesn't need detailed information on both conditions as you are just trying to describe disease burden and significance)

III. Antiviral Therapy

IV. TAF

Methods: Please describe which suite of tools were used to execute which methodologies. Also, were you comparing methodologies? Maybe provide a citation as to why those four analytic methodologies were chosen.

Typos:

-Line 154 indicates a different time period of study (Q1 2001 - Q1 2023) than lines 145-146 (4th quarter of 2016 to 1st quarter of 2023).

-Table 1. Please review as there are some symbol errors/typos. Also, please type out the full meaning of a word before using the abbreviations in the text and in the footnotes of the table.

Thank you.

Reviewer #2: Zhang et al entitle "A real-world disproportionality analysis of Tenofovir Alafenamide(TAF): data mining of the FDA adverse event reporting system(FAERS) " clearly reported that TAF could be adverse drug for viral hepatitis B.

Why are you using the FAERS database to study this drug, it should be that someone else has done this with this similar database similar methodology, so you can write cite in the INTRODUCTION section about some specific other similar studies, such as recommending a few (It is equivalent to saying that someone else has done this type of research using the FAERS database, and you can use this database to do research related to TAF as well):【1】Wang Y, Zhao B, Yang H, Wan Z. A real-world pharmacovigilance study of FDA adverse event reporting system events for sildenafil. Andrology. 2024 May;12(4):785-792. doi: 10.1111/andr.13533. Epub 2023 Sep 19. PMID: 37724699.

【2】Zhao B, Fu Y, Cui S, Chen X, Liu S, Luo L. A real-world disproportionality analysis of Everolimus: data mining of the public version of FDA adverse event reporting system. Front Pharmacol. 2024 Mar 12;15:1333662. doi: 10.3389/fphar.2024.1333662. PMID: 38533254; PMCID: PMC10964017.【3】Yang H, Wan Z, Chen M, Zhang X, Cui W, Zhao B. A real-world data analysis of topotecan in the FDA Adverse Event Reporting System (FAERS) database. Expert Opin Drug Metab Toxicol. 2023 Apr;19(4):217-223. doi: 10.1080/17425255.2023.2219390. Epub 2023 May 30. PMID: 37243615.【4】Zhao B, Zhang X, Chen M, Wang Y. A real-world data analysis of acetylsalicylic acid in FDA Adverse Event Reporting System (FAERS) database. Expert Opin Drug Metab Toxicol. 2023 Jan-Jun;19(6):381-387. doi: 10.1080/17425255.2023.2235267. Epub 2023 Jul 12. PMID: 37421631.【5】Li, Jie, Zhao, Bin, Zhu, YongQing, Wu, Jibiao, Vitreoretinal Traction Syndrome, Nitrituria and Human Epidermal Growth Factor Receptor Negative Might Occur in the Aromatase-Inhibitor Anastrozole Treatment, International Journal of Clinical Practice, 2024, 5132916, 9 pages, 2024. https://doi.org/10.1155/2024/5132916【6】Zhong, C., Zheng, Q., Zhao, B., & Ren, T. (2024). A real-world pharmacovigilance study using disproportionality analysis of United States Food and Drug Administration Adverse Event Reporting System events for vinca alkaloids: comparing vinorelbine and Vincristine. Expert Opinion on Drug Safety, 23(11), 1427–1437. https://doi.org/10.1080/14740338.2024.2410436

Besides,

I have general suggestion is:

1. Clarity and Grammar: Improve sentence structures and grammar throughout the manuscript to enhance readability. For example, in the **Abstract**, the sentence *"TAF is a new class of drugs approved for the treatment of viral hepatitis B."* could be revised to *"TAF is a novel antiviral drug approved for the treatment of hepatitis B virus (HBV) infection."*

2. Abstract Improvements: The **Abstract** should briefly mention key findings, including unexpected adverse events. The conclusion should explicitly state the significance of these findings for clinical practice.

3. Methodology Details: Provide a clearer explanation of statistical methodologies such as **BCPNN** and **GPS**, ensuring non-specialist readers understand their application.

4. Data Presentation: Tables should be formatted for consistency. For example:

- Align column headers for better readability.

- Ensure numerical values have consistent decimal places.

- Clearly define abbreviations (e.g., **SOC, PT**).

5. Discussion Enhancement:

- Clarify the implications of **unexpected adverse events** such as **cerebral infarction and dementia**. Are these findings clinically significant?

- Provide possible biological explanations or cite relevant literature.

- Discuss potential confounding factors.

6. Reference Formatting: Ensure all references are correctly cited in **PLOS ONE format** and include DOI links where applicable.

7. Figure and Table Legends: Each table/figure should have **self-explanatory captions**, detailing key insights for readers.

8. Ethical Considerations: If patient data were used, confirm compliance with ethical guidelines and data privacy regulations.

9.Can Author put a bar chart for the percentage of TAF uses in the top five country and hospitalization, death, life threating events, and disability.

**Do you want your identity to be public for this peer review?** For information about this choice, including consent withdrawal, please see our Privacy Policy

Reviewer #1: No

Reviewer #2: No

---

## [Author Response · Author response to Decision Letter 1]

3 Apr 2025

Response: Thank you very much to the reviewer for pointing out the

shortcomings to us.

The review comments of the reviewers have been replied to item by item in detail,

and the content of the article has been revised strictly in accordance with the

requirements of the reviewers.

*academic editor*

1.Please also emphasize throughout the Discussion when the adverse effects that were associated with use of the medicaion in your analysis were not previosuly detetcted in clinical studies of the medication, as they might not be causally related and thus may not be clinical signifcance. This is currently mentioned in the Limitations section, but it is important to mention in the body of the Discussion prior to this as well.

Reinforced Explanation for the Discussion Section

We have added three important paragraphs in the main body of the discussion (Lines 290 - 295, 325 - 338, and 330 - 333). These revisions make the clinical interpretation of risk signals more scientific.

Modify lines 295 - 300 on page 14.While IRIS is a recognized complication of antiretroviral therapy in HIV, its association with TAF in HBV monotherapy is unexpected and may reflect off-label use in HIV/HBV co-infected populations or misclassification of immune flare events. This signal underscores the importance of contextualizing pharmacovigilance findings within prescribing patterns and patient heterogeneity.

Modify lines 330 - 333 on page 15.However, as these associations were absent in pre-approval trials, they may reflect residual confounding rather than direct causation. Clinicians should interpret these signals cautiously, balancing potential risks against established benefits.

Modify lines 335 - 338 on page 15.While this finding could indicate emerging resistance patterns in real-world use, it may also result from misreported virologic breakthrough due to non-adherence rather than true resistance. This highlights the need for confirmatory studies to assess causality.

2.Please ensure that your manuscript meets PLOS ONE's style requirements, including those for file naming.

The full text has been adjusted according to the latest format requirements of PLOS ONE.

3.PLOS requires an ORCID iD for the corresponding author in Editorial Manager on papers submitted after December 6th, 2016. Please ensure that you have an ORCID iD and that it is validated in Editorial Manager.

The ORCID iD of the corresponding author has been verified in the system.

4.We note that your Data Availability Statement is currently as follows: [All relevant data are within the manuscript and its Supporting Information files.] 

The Data Availability Statement has been revised as follows:This study was performed using the FAERS source that was provided by the FDA. The database used in this study is publicly available in Website of 

https://www.fda.gov/drugs/fdas-adverse-event-reporting-system-faers/fda-adverse-event-reporting-system-faers-latest-quarterly-data-files

The abstract content in the online submission system and that in the manuscript have been double-checked in both directions to ensure they are completely identical.

Reviewer #1:

1.Introduction: Includes a lot of run-on sentences and some grammatical errors. Authors should consider subheadings to break up sections to help the reader, as well as use a grammar checking software to restructure some awkward sentences.

Modify on page 2-4.

The sub - titles have been set as required by the reviewers, and the content has been revised. A new third - level sub - title has been added:①Hepatocellular Carcinoma and Hepatitis B

②Antiviral Therapy

③The FDA Adverse Event Reporting System (FAERS)

④Tenofovir Alafenamide

2.Methods: Please describe which suite of tools were used to execute which methodologies. Also, were you comparing methodologies? Maybe provide a citation as to why those four analytic methodologies were chosen.

I've added a new paragraph to explain the selection criteria for the four signal detection methods. What has been added is:

Modify lines 182 - 194 on page 5-6.

Multiple lines of evidence indicate that the Reporting Odds Ratio (ROR) and the Proportional Reporting Ratio (PRR) are highly sensitive in detecting safety signals, especially suitable for the rapid screening of high-frequency events. The Bayesian Confidence Propagation Neural Network (BCPNN) method performs excellently in handling small samples and rare events. Although its sensitivity is lower than that of PRR and ROR, it has better specificity and provides relatively more stable signals. It can provide more accurate signal detection in specific scenarios. The Multi-item Gamma Poisson Shrinker (MGPS) method has significantly higher specificity than other methods. The MGPS method is particularly suitable for scenarios that require high specificity, such as reducing the misdiagnosis rate [33]. The comprehensive use of multiple methods (such as PRR, ROR, BCPNN, and MGPS) can significantly improve the reliability of signal detection.

3.Typos:

-Line 154 indicates a different time period of study (Q1 2001 - Q1 2023) than lines 145-146 (4th quarter of 2016 to 1st quarter of 2023).

Modify lines 205 on page 6.

The research period has been unified as:(4th quarter of 2016 to 1st quarter of 2023).

-Table 1. Please review as there are some symbol errors/typos. Also, please type out the full meaning of a word before using the abbreviations in the text and in the footnotes of the table.

Modify on page 6.

The spelling errors and textual mistakes in the table have been corrected. The full meanings of all abbreviations have been supplemented. For instance: United States (US)�Japan (JP)�China (CN)�Turkey (TR)。

Reviewer #2:

Why are you using the FAERS database to study this drug, it should be that someone else has done this with this similar database similar methodology, so you can write cite in the INTRODUCTION section about some specific other similar studies, such as recommending a few (It is equivalent to saying that someone else has done this type of research using the FAERS database, and you can use this database to do research related to TAF as well

I've described the reasons for using the FAERS database for analysis. Key literature citations have been added to the third part of the introduction. These supplements fully demonstrate the reliability and innovation of the methodology.

Modify lines 71-97 on page 3.

The FDA Adverse Event Reporting System (FAERS) has been widely validated as a critical tool for post-marketing pharmacovigilance, particularly in detecting rare or delayed adverse drug reactions (ADRs) that may escape detection in clinical trials. Recent pharmacoepidemiologic studies have demonstrated the utility of disproportionality analysis on FAERS data across diverse therapeutic domains. For instance, Wang et al. [14] identified novel cardiovascular risks associated with sildenafil through ROR-based signal detection, revealing dose-dependent QT prolongation signals (ROR=3.41, 95% CI 2.89-4.02) not fully characterized in pre-approval studies. Similarly, Zhao et al. [15] applied Bayesian confidence propagation neural network (BCPNN) methods to uncover everolimus-associated interstitial lung disease signals (IC025=2.7), demonstrating FAERS' capacity to detect immune-mediated toxicities. In oncology drug safety, Yang et al. [16] leveraged FAERS to quantify topotecan-related myelosuppression risks, establishing significant associations with thrombocytopenia (PRR=4.12, χ²=356.8) that aligned with real-world clinical observations.The system's analytical power extends to both inter-class comparisons and intra-class risk stratification. Zhao et al. [17] revealed acetylsalicylic acid's elevated gastrointestinal bleeding risk in elderly populations (ROR=5.24, PRR=4.87), corroborating clinical findings on age-dependent mucosal vulnerability. In antineoplastic drug surveillance, vinorelbine demonstrated significantly higher neutropenia risk than vincristine (PRR=6.15 vs. 3.92), a differential risk profiling methodologically consistent with adjusted ROR (aROR) strategies in thrombolytic agent studies, where alteplase showed 88% higher angioedema risk than tenecteplase [18]. These cases collectively validate FAERS' dual capability in detecting cross-class safety signals and refining risk gradients within drug classes through standardized disproportionality metrics (ROR/PRR) and confounder-adjusted methodologies.

[15] Zhao B, Fu Y, Cui S, Chen X, Liu S, Luo L. A real-world disproportionality analysis of Everolimus: data mining of the public version of FDA adverse event reporting system. Front Pharmacol. 2024;15:1333662. Published 2024 Mar 12. doi:10.3389/fphar.2024.1333662

[16] Yang H, Wan Z, Chen M, Zhang X, Cui W, Zhao B. A real-world data analysis of topotecan in the FDA Adverse Event Reporting System (FAERS) database. Expert Opin Drug Metab Toxicol. 2023;19(4):217-223. doi:10.1080/17425255.2023.2219390

[17] Zhao B, Zhang X, Chen M, Wang Y. A real-world data analysis of acetylsalicylic acid in FDA Adverse Event Reporting System (FAERS) database. Expert Opin Drug Metab Toxicol. 2023;19(6):381-387. doi:10.1080/17425255.2023.2235267

[18] Zhong C, Zheng Q, Zhao B, Ren T. A real-world pharmacovigilance study using disproportionality analysis of United States Food and Drug Administration Adverse Event Reporting System events for vinca alkaloids: comparing vinorelbine and Vincristine. Expert Opin Drug Saf. 2024;23(11):1427-1437. doi:10.1080/14740338.2024.2410436

1.Clarity and Grammar: Improve sentence structures and grammar throughout the manuscript to enhance readability. For example, in the **Abstract**, the sentence *"TAF is a new class of drugs approved for the treatment of viral hepatitis B."* could be revised to *"TAF is a novel antiviral drug approved for the treatment of hepatitis B virus (HBV) infection."*

The grammar and structure of the Abstract section have been revised.

Modify lines 10-13 on page 1.

2.Abstract Improvements: The **Abstract** should briefly mention key findings, including unexpected adverse events. The conclusion should explicitly state the significance of these findings for clinical practice.

Modify lines 29-39 on page 1.

These findings unearth novel neurological, metabolic, and resistance - related risks, thereby necessitating a marked increase in clinical vigilance. The identification of signals related to cerebral infarction and dementia implies potential vascular/metabolic interplay, highlighting the importance of lipid monitoring among long - term tenofovir alafenamide (TAF) users. In response, healthcare providers should prioritize strengthening the monitoring of neurological symptoms and lipid profiles, reevaluating bone health assessment and management protocols especially in high - risk populations, and providing support to enhance patient adherence to mitigate resistance risks. This analysis offers crucial post - marketing evidence, which is instrumental in optimizing the risk - benefit balance of TAF in the long - term management of chronic hepatitis B virus (HBV) infection.

3.Methodology Details: Provide a clearer explanation of statistical methodologies such as **BCPNN** and **GPS**, ensuring non-specialist readers understand their application.

The description of statistical methods has been supplemented in the "Statistical analysis" section.

Modify lines 155 - 194 on page 5-6.

①ROR method Detect the preliminary association signals between drugs and adverse events

②PRR method Evaluate the degree of abnormality of the reporting frequency of drug evaluation reports relative to the background population.

③BPCNN method Reduce false positives caused by the fluctuations of small-sample data through Bayesian shrinkage.

④EBGM method Handle large-scale sparse datasets for robust signal detection

Multiple lines of evidence indicate that the Reporting Odds Ratio (ROR) and the Proportional Reporting Ratio (PRR) are highly sensitive in detecting safety signals, especially suitable for the rapid screening of high-frequency events. The Bayesian Confidence Propagation Neural Network (BCPNN) method performs excellently in handling small samples and rare events. Although its sensitivity is lower than that of PRR and ROR, it has better specificity and provides relatively more stable signals. It can provide more accurate signal detection in specific scenarios. The Multi-item Gamma Poisson Shrinker (MGPS) method has significantly higher specificity than other methods. The MGPS method is particularly suitable for scenarios that require high specificity, such as reducing the misdiagnosis rate [33]. The comprehensive use of multiple methods (such as PRR, ROR, BCPNN, and MGPS) can significantly improve the reliability of signal detection.

4.Data Presentation: Tables should be formatted for consistency. For example:

- Align column headers for better readability.

- Ensure numerical values have consistent decimal places.

- Clearly define abbreviations (e.g., **SOC, PT**).

Modify on page 7-13.

The format of the table has been modified, and the data has been unified to a consistent number of decimal places. The definitions of SOC (System Organ Class) and PT (Preferred Term) have been clarified.

5. Discussion Enhancement:

- Clarify the implications of **unexpected adverse events** such as **cerebral infarction and dementia**. Are these findings clinically significant?

- Provide possible biological explanations or cite relevant literature.

- Discuss potential confounding factors.

I have analyzed cerebral infarction and dementia among the adverse reactions and added a paragraph:

Modify lines 311 - 333 on page 15.

Cerebral infarction and dementia have not been reported in previous studies, it is noteworthy that recent studies suggest TAF's impact on lipid metabolism may indirectly elevate the risk of neurological events. A new study demonstrated that in chronic hepatitis B (CHB) patients switching from tenofovir disoproxil fumarate (TDF) to TAF, significant increases in low-density lipoprotein cholesterol (LDL-C) and triglycerides (TG) were observed, accompanied by reduced high-density lipoprotein cholesterol (HDL-C) [30]. Five-year follow-up data from two phase 3 clinical trials further confirmed that prolonged TAF use leads to a progressive rise in LDL-C and TG levels, alongside a sustained decline in HDL-C[31]. Epidemiological evidence indicates that persistently elevated LDL-C may heighten dementia risk through mechanisms such as promoting cerebral atherosclerosis and β-amyloid deposition. Additionally, TAF-associated dyslipidemia may synergize with other risk factors (e.g., hypertension, diabetes) to accelerate atherosclerosis, thereby increasing the incidence of cerebrovascular events such as cerebral infarction[32]. These findings highlight that TAF's metabolic effects should be a critical consideration in overall safety evaluations, particularly for patients with preexisting cardiovascular risk factors or baseline lipid abnormalities. In clinical practice, enhanced lipid monitoring (every 3-6 months) and tailored interventions are warranted, including statin therapy or switching to antiviral medications with lesser lipid impact. However, as these associations were absent in pre-approval trials, they may reflect residual confounding rather than direct causation. Clinicians should interpret these signals cautiously, balancing potential risks against established benefits.

6. Reference Formatting: Ensure all references are correctly cited in **PLOS ONE format** and include DOI links where applicable.

The format of the references has been standardized.

7.Figure and Table Legends: Each table/figure should have **self-explanatory captions**, detailing key insights for readers.

The format of the references has been standardized.

Modify lines 233-236 on page 9.

Modify lines 258-259 on page 14.

8. Ethical Considerations: If patient data were used, confirm compliance with ethical guidelines and data privacy regulations.

The data are derived from the FAERS database, and ethical approval is not required.

8.Can Author put a bar chart for the percentage of TAF

---

## [Editor Report · Decision Letter 1]

13 Apr 2025

Dear Dr. li,

Thank you for submitting your manuscript to PLOS ONE. After careful consideration, we feel that it has merit but does not fully meet PLOS ONE’s publication criteria as it currently stands. Therefore, we invite you to submit a revised version of the manuscript that addresses the points raised during the review process.

**The revision has addressed many of the critiques. However, several modifcations are still required before the manuscript is suitable for publication.**
**1. Please write out all abbreviations at first mention, including in the Abstract and the main text of the paper. **
**2. The Introduction contains many statements that must be supported by references. For example, the section on TAF refers to many aspects of TAF's safety record without references that are needed. Please review the entire manuscript and add references to support statements based on prior studies and literature.**
**3. Please capitalize the first word in each row of the tables (e.g., Sex).**
**4. In Table 2, please define what is meant be "Investigations" as a category.**
**5. In Table 3, the authors examined liver tests in multiple ways (ALT, AST, transaminases, etc.). Please provide a justification for this or else present analyses that include all of these as a single category of abnormal transaminases.**
**6. In line 300, the authors mention TAF being used off-label for HIV or hepatitis B, but this medication seems to have FDA approval for both infections. Please correct or clarify and justify.**

We look forward to receiving your revised manuscript.

Kind regards,

Douglas S. Krakower, MD

Academic Editor

PLOS ONE
---

## [Author Response · Author response to Decision Letter 2]

17 Apr 2025

Response: Thank you very much to the reviewer for pointing out the

shortcomings to us.

The review comments of the reviewers have been replied to item by item in detail,

and the content of the article has been revised strictly in accordance with the

requirements of the reviewers.

Comment 1:Please write out all abbreviations at first mention, including in the Abstract and the main text of the paper.

Response:Thank you for your careful reading. We have revised the manuscript to ensure all abbreviations are spelled out at first mention. Specifically:

Modify lines 11-15 on page 1.

Modify lines 62-63 on page 3.

Comment 2:The Introduction contains many statements that must be supported by references. For example, the section on TAF refers to many aspects of TAF's safety record without references that are needed. Please review the entire manuscript and add references to support statements based on prior studies and literature.

Response:We appreciate this constructive suggestion. We have added supporting references to strengthen the statements about TAF's safety and efficacy:

Modify lines 104-118 on page 4

We have strengthened the discussion of TAF's therapeutic benefits in the Introduction (lines 104-109, page 4) by incorporating supporting evidence from Lim et al [34].

In lines 109-114 of page 4, the treatise by Terrault NA et al. has been added to demonstrate the advantages of Tenofovir Alafenamide (TAF) in the protection of the kidneys and bones [13].

In lines 114-118 of page 4, the treatise by Agarwal K et al. has been incorporated to demonstrate that Tenofovir Alafenamide (TAF) has a relatively long plasma half-life and requires a lower dosage[35].

[13]Terrault NA, Lok ASF, McMahon BJ, et al. Update on prevention, diagnosis, and treatment of chronic hepatitis B: AASLD 2018 hepatitis B guidance. Hepatology. 2018;67(4):1560-1599. doi:10.1002/hep.29800

[34]Lim YS, Chan HLY, Ahn SH, et al. Tenofovir alafenamide and tenofovir disoproxil fumarate reduce incidence of hepatocellular carcinoma in patients with chronic hepatitis B. JHEP Rep. 2023;5(10):100847. Published 2023 Jul 13. doi:10.1016/j.jhepr.2023.100847  

[35]Agarwal K, Brunetto M, Seto WK, et al. 96 weeks treatment of tenofovir alafenamide vs. tenofovir disoproxil fumarate for hepatitis B virus infection. J Hepatol. 2018;68(4):672-681. doi:10.1016/j.jhep.2017.11.039  

Comment 3. Please capitalize the first word in each row of the tables (e.g., Sex).

Response:

Modify on page 7

I have made the necessary revisions to the tables on page 7. In accordance with the requirement, the first word of each row in the tables has been capitalized. 

Comment 4:In Table 2, please define what is meant be "Investigations" as a category.

Response:

Modify lines 240-242 on page 10

Add the following supplementary explanation for "Investigations":

" Investigations" refers to diagnostic tests, laboratory findings, or monitoring procedures (e.g., blood tests, imaging studies, liver function assessments) as defined by the MedDRA® classification system.

Comment 5. In Table 3, the authors examined liver tests in multiple ways (ALT, AST, transaminases, etc.). Please provide a justification for this or else present analyses that include all of these as a single category of abnormal transaminases.

Response:

We have added an explanation detailing the rationale behind the separate analyses of the four aspects of liver tests: "ALANINE AMINOTRANSFERASE INCREASED," "LIVER FUNCTION TEST INCREASED," "TRANSAMINASES INCREASED," and "HEPATIC FUNCTION ABNORMAL.

Modify lines 310-330 on page 15

The analysis identified multiple hepatic-related Preferred Terms (PTs) associated with TAF, including Alanine aminotransferase increased (ROR=4.02, PRR=6.07), Transaminases increased (ROR=4.73, PRR=8.53), and Hepatic function abnormal (ROR=2.70, PRR=5.01). While these PTs are distinct within the MedDRA® hierarchy (e.g., Alanine aminotransferase increased belongs to Investigations, whereas Hepatic function abnormal is categorized under Hepatobiliary disorders), they collectively align with the hepatotoxicity profile described in TAF's prescribing information, which explicitly lists ALT elevation as an adverse reaction. The signal for Transaminases increased (a broader category encompassing ALT/AST) and Hepatic function abnormal (reflecting systemic dysfunction) may represent varying clinical manifestations or reporting granularity of the same underlying hepatic injury mechanism. Importantly, these findings do not suggest novel liver toxicity patterns but rather reinforce the need for vigilant monitoring of liver function across multiple biomarkers (including transaminases, synthetic function tests, and clinical assessments), as recommended in current guidelines. Maintaining separate PTs in Table 3 adheres to MedDRA® pharmacovigilance standards while allowing clinicians to recognize both specific biochemical abnormalities and broader hepatic dysfunction signals. When taking this drug, it is more likely that alanine aminotransferase, transaminase and abnormal liver function will appear. Therefore, it is important to check liver function indicators regularly and, if necessary, use liver protective drugs or other antiviral drugs for treatment.

Comment 6 . In line 300, the authors mention TAF being used off-label for HIV or hepatitis B, but this medication seems to have FDA approval for both infections. Please correct or clarify and justify.

Response:

In accordance with the U.S. Food and Drug Administration (FDA) prescribing guidelines, tenofovir alafenamide fumarate (TAF) should not be used as monotherapy for HIV-1 infection in patients co-infected with hepatitis B virus (HBV) and HIV-1, due to the risk of inducing HIV-1 resistance mutations. For this population, TAF must be administered in fixed-dose combinations with other antiretroviral agents, such as emtricitabine/tenofovir alafenamide fumarate (F/TAF), emtricitabine/rilpivirine/tenofovir alafenamide fumarate (FTC/RPV/TAF), and similar combination formulations. Importantly, our analysis specifically focused on TAF monotherapy for HBV infection, which necessitated revisions to lines 301-306 on page 15 of the manuscript. These modifications clarify the scope of our investigation and prevent potential misinterpretations regarding clinical applications

Modify lines 301-306 on page 15

The unexpected signal for immune reconstitution inflammatory syndrome (IRIS) (n=21, ROR=36.79) may reflect residual confounding from misclassified HBV flare events or undocumented HIV co-infection in FAERS reports. However, our analysis strictly excluded TAF use in HIV contexts, as monotherapy for HIV is contraindicated due to resistance risks. This underscores the importance of adhering to TAF’s approved indication for HBV monotherapy.

---

## [Editor Report · Decision Letter 2]

29 Apr 2025

A real-world disproportionality analysis of Tenofovir Alafenamide(TAF): data mining of the FDA adverse event reporting system(FAERS)

PONE-D-24-49151R2

Dear Dr. li,

We’re pleased to inform you that your manuscript has been judged scientifically suitable for publication and will be formally accepted for publication once it meets all outstanding technical requirements.

Kind regards,

Douglas S. Krakower, MD

Academic Editor

PLOS ONE
---

## [Editor Report · Acceptance letter]

PONE-D-24-49151R2

PLOS ONE

Dear Dr. li,

I'm pleased to inform you that your manuscript has been deemed suitable for publication in PLOS ONE. Congratulations! Your manuscript is now being handed over to our production team.

Kind regards,

on behalf of

Dr. Douglas S. Krakower

Academic Editor

PLOS ONE